# Enhancing X-ray-Based Wrist Fracture Diagnosis Using HyperColumn-Convolutional Block Attention Module

**DOI:** 10.3390/diagnostics13182927

**Published:** 2023-09-13

**Authors:** Joonho Oh, Sangwon Hwang, Joong Lee

**Affiliations:** 1Department of Mechanical Engineering, Chosun University, Gwangju 61452, Republic of Korea; hi990@naver.com; 2Department of Precision Medicine, Yonsei University Wonju College of Medicine, Wonju 26426, Republic of Korea; arsenal@yonsei.ac.kr; 3Artificial Intelligence BigData Medical Center, Yonsei University Wonju College of Medicine, Wonju 26426, Republic of Korea

**Keywords:** artificial intelligence, biomedical image processing, bone fractures, HyperColumn, wrist fractures, X-ray

## Abstract

Fractures affect nearly 9.45% of the South Korean population, with radiography being the primary diagnostic tool. This research employs a machine-learning methodology that integrates HyperColumn techniques with the convolutional block attention module (CBAM) to enhance fracture detection in X-ray radiographs. Utilizing the EfficientNet-B0 and DenseNet169 models bolstered by the HyperColumn and the CBAM, distinct improvements in fracture site prediction emerge. Significantly, when HyperColumn and CBAM integration is applied, both DenseNet169 and EfficientNet-B0 showed noteworthy accuracy improvements, with increases of approximately 0.69% and 0.70%, respectively. The HyperColumn-CBAM-DenseNet169 model particularly stood out, registering an uplift in the AUC score from 0.8778 to 0.9145. The incorporation of Grad-CAM technology refined the heatmap’s focus, achieving alignment with expert-recognized fracture sites and alleviating the deep-learning challenge of heavy reliance on bounding box annotations. This innovative approach signifies potential strides in streamlining training processes and augmenting diagnostic precision in fracture detection.

## 1. Introduction

According to the Korea Health Insurance Review and Assessment Service in 2022, there were 4,881,990 patients diagnosed with fractures, of which 884,296 were hospitalized due to the severity of their injuries. This represents approximately 9.45% of South Korea’s population, which stood at 51.63 million in the same year, emphasizing the high prevalence of fractures within the nation. While specific data on fractures in the upper extremities are not available for this year, it is noteworthy that in 2020, around 25% of fracture cases were related to the upper extremities [1].

X-ray imaging was first introduced in the Korean Government-General of Chosen hospitals in 1911. Over a century later, it still serves as the primary tool for fracture assessments, especially during initial evaluations in emergency departments. While advanced imaging modalities like CT and MRI are not commonly found in primary care centers due to cost considerations, X-ray remains the go-to diagnostic tool. However, interpreting X-ray images can be challenging, even for seasoned physicians. Missing a fracture on an X-ray can lead to grave consequences for the patient, including delayed treatment and hindered functional recovery. Furthermore, radiographs in emergency settings are frequently read by emergency medicine clinicians who may lack in-depth orthopedic expertise. Fracture detection can be especially difficult with simple X-rays, as opposed to CT or MRI, particularly in complex anatomical structures like the carpal bones and minor intra-articular impingement fractures of the fingers. This underlines the necessity for the development and application of computer-aided diagnosis (CAD) systems [2,3,4,5,6].

Recent advances in machine learning, particularly deep-learning architectures, have demonstrated considerable potential in image analysis. This technology has shown promise across various sectors of computer vision and has made notable strides in medical diagnostics. Andre Esteva et al. demonstrated dermatologist-level performance in identifying skin cancer using a convolutional neural network (CNN) model [7], Inception-v3, on a large skin cancer image dataset. Gulshan et al. outperformed the average human ophthalmologist in detecting diabetic retinopathy using Inception-v3 [8]. Pranav Rajpurkar et al. achieved radiologist-level performance on 14 diseases using chest X-ray images and a 121-layer CNN model called CheXNet [9].

Recent technological advancements have ushered in a new era for fracture diagnosis using radiography. Although previous research has delved into this area, our study introduces a novel combination of HyperColumn techniques and the convolutional block attention module (CBAM) to significantly enhance fracture detection in radiographs [10]. Unlike previous methodologies, this approach amalgamates features from different resolutions. A distinctive feature of this research is the employment of a model trained exclusively with fracture/non-fracture labelled data, obviating the need for bounding box annotations.

The main contributions of this study are as follows: (1) The comparison of basic models for classifying wrist bone X-ray images as either fracture or non-fracture. Ten architectures were trained using cutting-edge data-augmentation techniques, such as mix-up augmentation, and the results were compared. (2) The improvement in the classification results by employing HyperColumn and attention mechanisms in the base model. (3) Analysis of the network’s regions of interest by extracting heatmaps using GRAD-CAM technology and demonstration of how the HyperColumn and attention significantly boost the accuracy of fracture prediction. (4) The increase in fracture prediction by incorporating coordinate information into the proposed network. (5) The demonstration of the proposed HyperColumn-CBAM model’s generalizability and scalability, which shows predictions similar to those of fracture regions marked by experts, without bounding box annotation.

In the midst of burgeoning advancements in medical imaging and diagnostics, computational simulations or in silico studies have risen as a formidable avenue for investigation. With benefits such as cost-efficiency and rapid results, they are fast becoming a potent counterpart to traditional in vivo and in vitro studies. The implications of these simulations, especially in the realm of fracture diagnostics, are gaining momentum. Our methodology contributes to harnessing this potential, ushering in a new epoch for medical diagnostics [11].

The remainder of the manuscript is organized as follows: Section 2 discusses the related research. Section 3 details the materials used, specifically the database of radiographs. Section 4 describes the methods employed, including deep-learning techniques, along with performance metrics such as HyperColumn, CBAM, accuracy, and Cohen’s kappa coefficient. Section 5 and Section 6 present the experimental results and discussions, respectively. The manuscript concludes in Section 6 with a discussion of the results.

## 2. Related Works

In the continuously evolving domain of medical imaging and deep learning, new applications are consistently emerging. For instance, Brederoo et al. underscored the potential of automatic speech analysis for the early detection of psychiatric symptoms [12]. Yildirim et al. leveraged MFCC and CNN for heart sound analysis, attaining a diagnostic accuracy of 99.5% in identifying heart valve diseases [13]. In the realm of Alzheimer’s disease diagnosis using MRI, AlSaeed et al. harnessed the capabilities of convolutional neural networks (CNNs) to enhance and expedite the diagnostic procedure [14].

Additionally, the progress in CNN-based studies for identifying and classifying fractures, especially in the upper extremities, is noteworthy. The following highlights some of the notable contributions. Below are general studies on upper extremities. Chung et al. reported an accuracy of 96% using a ResNet-152 CNN model trained on 1891 shoulder radiographs [15]. Olczak et al. evaluated five models with 256,000 radiographs of the wrist, hand, and ankle, finding that the VGG16 network had the highest accuracy of 93% [16]. Rayan et al. trained Xception + RNN with 58,817 elbow radiographs from 21,456 children, achieving an accuracy of 88% and an AUC of 0.95 [17]. Pranav Rajpurkar et al. trained the DenseNet169 model on 40,561 upper-extremity radiographs from 12,173 patients and achieved an AUC of 0.929, similar to the accuracy of radiologists. They also released the MURA dataset for research purposes [18]. Ghoti et al. utilized the MURA dataset and trained it using U-Net, achieving a Dice coefficient of 90.29% for fracture segmentation [19]. Tabarestani et al. reported that Faster-RCNN, using X-ray images of the humerus, elbow, and forearm from the MURA database, showed an average precision (AP) of 66.82% (Intersection over Union (IOU) = 50%) [20].

While some studies have considered the entire upper extremity, including wrist fractures, others have focused solely on wrist fractures. Kim et al. evaluated the Inception V3 model with 1112 wrist radiographs images for training and 100 testing images, reporting an AUROC of 0.954 [21]. Lindsey et al. trained U-Net on 34,990 wrist radiographs, reporting an AUC of 0.967. They claimed that their method improved clinicians’ image reading from 88% to 94% (unaided and aided, respectively), with a 53% reduction in misinterpretations [22]. Ananda et al. compared 11 CNN networks in classifying wrist radiographs as normal or abnormal using the MURA dataset, finding Inception-ResNet-v2 to have the highest accuracy of 86.9% [23]. Nagy et al. reported that Yolo5, trained on 74,459 annotations from 20,327 pediatric trauma wrist radiographs for 6091 patients, achieved a precision of 0.917, a recall of 0.887, and a mean average precision (mAP) of 0.933 (based on an IoU threshold of 0.5). They also released the GRAZPEDWRI-DX dataset for research purposes [24]. Ju et al. used the GRAZPEDWRI-DX dataset for wrist radiographs with data augmentation and trained the YOLOv8 model, reporting a mean average precision (mAP 50) of 0.631 at an input image size of 1024 [25].

On a broader scale, innovations in fracture detection are ongoing. Ma et al. employed Faster R-CNN to identify 20 bone regions in X-ray images and subsequently optimized accuracy through a two-stage approach featuring the convolutional neural network, Crack-Net [26]. Meanwhile, Moon et al. achieved a mean average precision (mAP) of 69.8% across diverse facial fracture datasets using the YoloX-S object-detection model with CT images, which inherently provide richer information compared to X-rays [27]. Venturing into the 3D imaging space, Yang et al. utilized generative adversarial networks (GANs) to extrapolate 3D spinal structures from 2D X-ray images, leveraging a dataset of 1012 CT images from 984 patients [28].

These advances are promising for wrist fracture diagnosis. Strategies that blend generative adversarial networks (GANs), hierarchical CNNs, and diverse neural architectures—especially those for data augmentation—and incorporate insights from auxiliary datasets, such as CT scans, are likely to be pivotal in this domain.

## 3. Materials

The musculoskeletal radiographs (MURA) dataset is a large compilation of musculoskeletal radiographs for seven upper extremities: the elbow, finger, forearm, hand, humerus, shoulder, and wrist. These were obtained from 12,173 patients at Stanford Hospital between 2001 and 2012 and were manually classified as normal or abnormal by Stanford Hospital radiologists. The total number of images is 40,561, of which 9752 (negative: 5765, positive: 3987) are wrist images. In addition, the validation dataset comprises 659 images (negative: 364, positive: 259). Stanford University has released the MURA dataset for research purposes. Figure 1 provides examples from the positive dataset.

The GRAZPEDWRI-DX dataset consists of 6091 pediatric trauma wrist radiographs treated at the Department of Pediatric Surgery, University Hospital Graz, between 2008 and 2018. In total, 74,459 image tags were assigned to the 20,327 images. These tags included bone anomaly, bone lesion, foreign body, fracture, metal, periosteal reaction, pronator sign, soft tissue, text, and centerline. Of these, 18,090 boxes were marked as fractures, and 13,550 images had one or more fracture boxes. These were reviewed at least twice by a pediatric radiologist. The authors made the data publicly available. Figure 2 shows an example of multiple bounding boxes in one of the GRAZPEDWRI-DX datasets.

Although other publicly available datasets for fractures exist, only the wrist bone set from the MURA dataset was utilized out of the seven types of MURA data. This choice was influenced by the fact that the extensive GRAZPEDWRI-DX dataset comprises solely wrist bones, aligning with the objective to employ both datasets concurrently. Furthermore, for classification, as a dataset consisting only of fractures/non-fractures without bounding boxes, visualization of the trained network, and calculation of the concordance rate between the fracture sites predicted by the network and the fracture sites marked by orthopedic surgeons, the wrist sites shared by the two sets are optimal. However, while the MURA dataset is open source, only the training and validation datasets are publicly available. The study conducted in the competition was tested using testing data that are not publicly available. Due to the confidentiality of the MURA testing data, reproducing the results with the same dataset is not feasible. For Table 1, the MURA validation data were employed as testing data for fracture classification in this study. Moreover, given the absence of designated testing and validation sets in the GRAZPEDWRI-DX dataset, 20% of its total images, combined with the MURA validation set, were used to determine the accuracy of the fracture box.

## 4. Methods

Deep learning has catalyzed the development of numerous network architectures, each intricately designed to address specific image sizes, computational demands, and distinct goals. Amidst this extensive array, this study harnesses several pioneering and influential architectures. Various deep-learning methodologies were applied to categorize X-ray images of wrist bones into fractures and non-fractures and to further pinpoint the exact fracture sites in images flagged for fractures. The approach began with a classification exercise employing eight CNN architectures and two transformer models, as enumerated in Table 1. Informed by the results obtained, a novel network was formulated. Going beyond just DenseNet169 and EfficientNet-B0, both of which exhibited outstanding classification capabilities, the integration of the HyperColumn and attention modules was introduced to enhance the efficacy of the classification and fracture-site-prediction tasks.

VGG19: This model is equipped with 16 convolutional layers, complemented by 3 fully connected layers. Upon its debut, its profound depth enabled it to adeptly discern complex semantics, heralding a new era in image classification [29].

ResNet50: This architecture is adorned with 50 layers, 49 of which are convolutional, with 1 being fully connected. ResNet50’s hallmark lies in its four residual blocks, ingeniously designed to combat the infamous vanishing gradient issue in deep networks, thereby preserving a coherent information flow [30].

Inception v3: With its intricate arrangement of 48 convolutional layers and a global average pooling layer, Inception v3 champions the use of factorized convolutional decomposition. This finesse ensures optimal computational efficiency, delivering a performance on par with VGGNet but in a more parameter-efficient manner [31].

Xception: An evolution of the Inception model, Xception boasts 71 convolutional layers coupled with a global average pooling layer. It sets itself apart by discerning channel-wise correlations and spatial information in images independently, resulting in amplified performance [32].

RegNet-064: Conceived by Radosavovic et al., RegNet-064 embodies a refined design ethos, showcasing regularized network paradigms. The design objective is clear: create architectures that are both uniformly structured and inherently intuitive, all while upholding stellar performance [33].

NASNet: Emerging from the innovative neural architecture search (NAS) paradigm proposed by Zoph et al., NASNet diverges from traditionally curated models like ResNet and Inception. Eschewing manual design, NASNet’s blueprint stems from autonomously forged convolutional blocks. This avant-garde approach amalgamates reinforcement learning and RNNs, weaving the unique NASNet fabric [34].

Vit_base_patch16_224: Crafted by Dosovitskiy et al., this model repurposes transformer mechanisms—originally the stalwarts of natural language processing (NLP)—for image classification. By viewing image patches as sequential entities, the model attests to the versatility of transformers, proving their prowess extends beyond text to visual domains. This novel strategy frequently eclipses conventional CNN architectures, all while being more resource-efficient during training [35].

Swin Transformer: An innovation by Liu et al., the Swin Transformer reimagines the transformer design for vision-centric applications. Drawing inspiration from the Vision Transformer, it dissects images into discrete non-overlapping patches. Its characteristic lies in the incorporation of a dynamic window mechanism, astutely discerning both granular and global visual cues. This approach has solidified the Swin Transformer’s dominance across an array of vision benchmarks, spanning image classification, object detection, and semantic segmentation [36]. Table 1 presents an overview of various deep-learning network models used in our study.

### 4.1. Base Models

EfficientNet

Three main strategies exist to enhance model performance. First, adding more layers to the model allows it to learn more complex and abstract features from the input data. Second, increasing the number of filters in the convolutional layer enables the model to extract finer-grained features. Lastly, high-resolution input images provide the model with more detailed visual information, facilitating more accurate predictions. EfficientNet employs automated machine learning (AutoML) to determine the optimal combination of these three scaling factors. The authors proposed a composite scaling method that efficiently integrates these factors. Utilizing the MBConv block, a convolutional block inspired by MnasNet, and incorporating squeeze and excitation optimization techniques, they managed to achieve state-of-the-art (SOTA) performance with a relatively small model. EfficientNet is available in various versions, ranging from B0 to B7 [37].

DenseNet

In DenseNet, the feature maps of all layers are concatenated, such that each layer influences every other layer. Unlike ResNet, which employs addition, DenseNet uses concatenation. This approach mitigates the vanishing gradient problem, enhances feature propagation, and encourages feature reuse. Additionally, DenseNet requires fewer parameters than traditional convolutional networks because it does not need to relearn replicated feature maps [38].

### 4.2. CBAM Module

The convolutional block Attention nodule (CBAM) comprises a channel attention module and a spatial attention module, as depicted in Figure 3b Each attention module generates an attention map for the channel and space. These attention maps suppress unnecessary information and emphasize important information. This information is then used to resize the feature map, giving more importance to more relevant locations [39].

Channel Attention Module

As depicted in Figure 3a, the feature map obtained from the previous convolution layer, denoted by *F*, is initially processed to create features that exclusively contain channel information, with the spatial information being removed through global average pooling and max pooling. The two resultant vectors are then passed through a multilayer perceptron (MLP) to introduce nonlinearity. The MLP reduces the dimensions by a ratio of C/r, with a hidden layer in between. r is chosen to be 16. When the input F is C × H × W, the outcome of the channel attention is C × 1 × 1. The channel attention module is computed as per Equation (1) Here, σ denotes a sigmoid function, and W_1_ and W_0_ are the weights of the MLP.
M_c_(F) = *σ*(MLP(AvgPool(F)) + MLP(MaxPool(F)))= *σ*(W_1_(W_0_(F*^c^_Avg_*)) *+ σ*(W_1_(W_0_(F*^c^_Max_*))))(1)

Spatial Attention Module

The spatial attention module focuses on identifying the areas containing essential information. As depicted in Figure 3c, it concatenates two values, *F_avg* and *F_max*, both of which are 1 × H × W. These values are generated by applying Maxpool and *Avgpool* operations to *F*, which are created by multiplying the channel attention map with the input feature map along the channel axis. Then, it applies a 7 × 7 convolution operation to generate a spatial attention map. The spatial attention module is computed as per Equation (2).
M_s_(F) = *σ*(f^7×7^([AvgPool(F); MLP(MaxPool(F)))
= *σ*(f^7×7^ ([F*^c^_Avg_;* F*^c^_Max_*]))(2)

### 4.3. Proposed Method

Methods that employ convolutional neural networks (CNNs) for image classification can face challenges when the object of interest, such as a fracture, constitutes only a small portion of the image. This difficulty stems from the network’s tendency to average the features across the entire image space. In these instances, another deep-learning approach, namely object detection, might prove advantageous. However, many fracture datasets, including the MURA dataset, do not furnish bounding box annotations. This lack of bounding box information arises because these datasets classify images as either fracture or non-fracture without bounding boxes, as the process of annotating bounding boxes is time-consuming and requires specialist expertise. To circumvent this issue, the HyperColumn-CBAM module was introduced.

In a standard CNN, the feature-extraction process comprises several convolutional layers, followed by downsampling operations, such as pooling or stride convolution. Regrettably, this downsampling often leads to a loss in spatial resolution, thereby causing the loss of crucial details associated with specific object features or local characteristics. The HyperColumn technique was introduced to alleviate these limitations. As depicted in Figure 4, the HyperColumn technique upsamples the feature maps of the downsampled layers and stacks them to form a HyperColumn feature map. This map retains both low-level and high-level information, where lower-level information conveys fine-grained detail, and higher-level information encapsulates more abstract global features, offering the advantage of capturing both intricate details and broader patterns [40,41].

Nonetheless, there is a risk that less useful features may overwhelm the more salient ones for classification when features extracted at various resolutions are combined without consideration of their relative importance. To address this, a model is proposed that assigns weights to the HyperColumn extracted at different resolutions based on the importance of spatial features, instead of directly utilizing them.

DenseNet169 and EfficientNet-B0 were utilized as base models, both of which have demonstrated high performance in classification tasks. As illustrated in Figure 5, for EfficientNet-B0, downsampling occurs in the initial convolution and in the MBConv block where the stride is set to 2. In this case, the CBAM, comprising both channel and spatial attention modules, is used on the feature maps extracted from all when a decrease in resolution occurs. As shown in Figure 6, for DenseNet169, the feature map was extracted from the last convolution performed before downsampling took place, and we applied a CBAM that assigned weights to the channels based on their spatial and channel importance. Subsequently, global mean pooling and global maximum pooling were applied to each feature image, thereby compressing them with minimal information loss. Lastly, a fully connected layer and a sigmoid layer were then utilized for fracture classification.

This approach ensures that the heatmap extracted for fine features, such as fractures, accurately represents the overall shape, thus facilitating a high-quality classification performance.

### 4.4. Extracting Heatmaps and Predicting Bounding Boxes

Gradient-weighted class activation mapping (Grad-CAM) was utilized to identify the network’s area of focus and predict the fracture site accordingly. The gradient of the class score was computed with respect to the feature maps in the final convolutional layer. Each feature map was then weighted based on the gradient’s strength, and these weighted feature maps were summed to generate a heatmap. The resulting heatmap was normalized to a range between 0 (representing lows) and 255 (representing highs) using the peaks and troughs and was visualized as a jet color map, in which blue indicates the lowest activation, and green to red represents increasingly higher activation levels. Squares with an activation value above a certain threshold (230 in our case) were marked as potentially indicating fractures. These were compared to the squares marking the fracture site as annotated by the medical expert. As depicted in Figure 7, the far-left image is the input fracture image, the middle image overlays the original image with the heatmap, and the far-right image shows the predicted fracture site based on our threshold criterion.

### 4.5. Preprocessing and Data Augmentation

MURA X-ray images are often characterized by dark backgrounds and a lack of contrast. Furthermore, the area of interest frequently occupies only a portion of the image, leaving much of the image space empty. Conversely, GRAZPEDWRI-DX images are fully occupied and display higher sharpness. To achieve sharper images, the contrast-limited adaptive histogram equalization (CLAHE) technique was employed. This technique enhances image contrast by adjusting the histogram of each section of the image based on a user-defined clip limit. Given that the image sharpness depends on the clip limit, four sharpened renditions of each image were produced using different clip limits, in addition to the original image. This process resulted in five-channel images. The CLAHE transformed image was also used to apply Otsu’s thresholding method to separate the bones from the background, and then we rescaled the bone images to fill the entire frame.

For enhanced performance, several data-augmentation techniques were employed: rotation, horizontal flipping, color inversion, and mix-up augmentation, all applied within a range of −30 to 30 degrees to every image in the training set. Mix-up augmentation creates synthetic examples in the input space by performing linear interpolation between random pairs of training examples (encompassing both inputs and labels). This method results in smoother decision boundaries for the model, reducing the likelihood of overfitting. An alpha value of 0.01 was used for mix-up augmentation [42].

### 4.6. Evaluation Metrics

To assess the performance of our proposed method, the following evaluation metrics were employed: accuracy, precision, sensitivity, F1-score, Cohen’s kappa score, and IoU (Intersection over Union for object detection), as defined in Equations (3)–(8) [43,44,45].

Accuracy represents the proportion of correct predictions of the total data:(3)$$\mathrm{Accuracy}=\frac{\left(\mathrm{TP}\;+\;\mathrm{TN}\right)}{\;\left(\mathrm{TP}\;+\;\mathrm{TN}\;+\;\mathrm{FP}\;+\;\mathrm{FN}\right)}$$

Precision is the proportion of positive predictions that are actually positive:(4)$$\mathrm{Precision}=\frac{\mathrm{TP}}{\;\left(\mathrm{TP}\;+\;\mathrm{FP}\right)}$$

Sensitivity, or recall, represents the proportion of positives that are correctly identified:(5)$$\mathrm{Sensitivity}=\frac{\mathrm{TP}}{\;\left(\mathrm{TP}\;+\;\mathrm{FN}\right)}$$

The F1 score is the harmonic mean of precision and recall:(6)$$F1\;\mathrm{score}=\frac{2\times \left(\mathrm{Precision}\;\times \;\mathrm{Sensitivity}\right)}{\left(\mathrm{Precision}\;+\;\mathrm{Sensitivity}\right)}$$

Cohen’s kappa (κ) index is a statistical measure of agreement that accounts for the probability of raters classifying material into the same category by chance. If *P*_0_ represents the observed agreement proportion, and *Pc* denotes the expected proportion of agreement by chance, the kappa statistic is defined as the ratio of the difference between the observed agreement and the agreement due to chance to the difference between perfect agreement and the agreement due to chance (Equation (7)): [46]
(7)$$k=\frac{P_{0}-P_{C}}{1-P_{C}}$$

Intersection over Union (IoU) is a measure ranging from 0 to 1 that quantifies the overlap between the ground truth and the predicted bounding box. If two equal-sized squares are offset by one-ninth, the IoU will be approximately 0.65. This measure is used to gauge the accuracy of the predicted bounding box [47].
(8)$$\mathrm{IoU}=\frac{\mathrm{Area}\;\mathrm{of}\;\mathrm{Overlap}}{\mathrm{Area}\;\mathrm{of}\;\mathrm{Union}}$$

The receiver operating characteristic (ROC) curve illustrates the performance of a binary classifier as it navigates through various thresholds. It is a curve that demonstrates how performance metrics alter as the threshold changes, with the true positive rate (Recall) plotted on the y-axis and the false positive rate (1-Specificity) on the x-axis.

The area under the curve (AUC) value is the region beneath the ROC curve, and it typically yields better results when it is closer to 1. Achieving a large AUC relies heavily on the ability to obtain a high true positive rate while maintaining a low false positive rate. Therefore, the closer the ROC curve shifts away from the diagonal line (representing a random classifier) and towards the upper left corner (indicating a perfect classifier), the closer the AUC is to 1, signifying an outstanding model performance [48].

As a rule of thumb:−An AUC of 0.5 suggests the classifier performs no better than random chance.−An AUC in the range of 0.7 to 0.8 is deemed acceptable.−An AUC in the range of 0.8 to 0.9 is considered excellent.−An AUC of 0.9 or above is considered outstanding.

## 5. Results

Each of the models outlined in Table 1, including the two proposed HyperColumn-CBAM models, were trained, and their performances were evaluated. The data consisted of X-ray images of wrist bones from the MURA dataset. The preprocessing and augmentation techniques detailed in the previous section were employed, setting the number of epochs to 120 and the batch size to 32, and using the Lion optimizer and binary weighted cross-entropy loss. The initial learning rate was set to 0.0001 but was not fixed; it was decreased by a factor of 10 if the loss did not improve after 10 iterations to boost the network’s learning efficiency.

Table 2 presents the accuracy, precision, recall, F1 score, and Cohen’s kappa score obtained for each trained model during classification on the testing set. EfficientNet-B0 exhibited the highest training accuracy without the use of HyperColumn. Moreover, both DenseNet169 and EfficientNet-B0, which served as base models, displayed improvements across all evaluation metrics, including accuracy and precision, when the HyperColumn-CBAM architecture was integrated. Interestingly, both DenseNet169 and EfficientNet-B0 performed similarly when employing the HyperColumn-CBAM structure, but DenseNet169 demonstrated a slight overall advantage. For EfficientNet-B0, the accuracy improved by approximately 0.69%, while Cohen’s kappa score rose by 1.64%. For DenseNet169, the accuracy increased by about 0.70%, with the Cohen’s kappa score rising by 5.07%.

Figure 8 presents the fracture classification results using the DenseNet169 and EfficientNet-B0 models, alongside the outcomes of enhancing these models with HyperColumn-CBAM, using the AUC scores and ROC curves as measures. The HyperColumnCBAM-DenseNet169 model exhibited the best predictive performance with an AUC of 0.9145. Notably, augmenting the DenseNet169 model with HyperColumn-CBAM enhanced the AUC score from 0.8778 to 0.9145. Similarly, the EfficientNet-B0 model experienced an AUC increase from 0.8937 to 0.9071 upon augmentation with the HyperColumn-CBAM model. Figure 9 presents the confusion matrix for HyperColumn-CBAM-DenseNet169 and HyperColumn-CBAM-EfficientNet-B0.

Table 3 displays the performance of DenseNet169 on wrist bone X-ray images from both the MURA dataset and images with fracture boxes from the GRAZPEDWRI-DX set. For this, 80% of the total images served as the training set and 20% served as the testing set. As the MURA dataset lacks bounding boxes, while the GRAZPEDWRI-DX set contains them, the metrics of accuracy, precision, recall, F1, and Cohen’s kappa score were calculated using the testing set premise. However, for the IoU calculation, only the GRAZPEDWRI-DX images in the testing set with bounding boxes were used. The model using HyperColumn-CBAM outperformed the default DenseNet169 on all the evaluation indices, including accuracy. Furthermore, the HyperColumn-CBAM IoU significantly increased from 0.083 to 0.213. The results of the simultaneous classification and coordinate estimation were examined by adding four inputs to the HyperColumn-CBAM model. For fracture box estimation, direct coordinate regression with Smooth L1 Loss was used instead of the heatmap method. Multitasking resulted in a slightly lower accuracy compared to fracture classification alone but improved the IoU from 0.213 to 0.574.

To understand and analyze the learned features of the model, Grad-CAM analysis was conducted on the final convolutional channel using the testing set images. Figure 10 presents the results for four samples from the testing set. Notably, the heatmaps of the model enhanced with HyperColumn-CBAM more consistently highlighted areas with concentrated fractures in the wrist than those obtained with DenseNet169. For the DenseNet169 model, the maximum heatmap area was often located closer to the fracture site than to the pseudo-fracture site, or the heatmap appeared wider. Conversely, for the HyperColumn-CBAM-DenseNet169 model, the maximum heatmap area was located closer to the fracture site and was narrower and thus closer to the area marked by the specialist. It was also observed that the HyperColumn-CBAM-DenseNet169 model, trained by adding coordinate information, was the closest to the area marked by the specialist, which is corroborated by the actual image in terms of the IoU.

## 6. Discussion

In this study, data categorized as fracture/non-fracture in wrist X-rays were analyzed. Our focus was on (1) the difference in prediction performance achieved by training with ten distinct pretrained deep-learning models, (2) the ability to predict the fracture site using the heatmap of the model trained for fracture classification, and (3) whether the HyperColumn-CBAM structure enhanced the prediction performance of the deep-learning model and the fracture site estimation derived from the heatmap.

Recently, convolutional neural networks (CNNs) have been predominantly used for image analysis. However, the Vision Transformer (ViT) family has significantly improved its performance, prompting us to train eight CNNs and two ViT family models for wrist X-ray classification and evaluate their effectiveness.

This study pioneers the use of the HyperColumn-CBAM structure to enhance the prediction performance of existing models on wrist X-ray images.

Among these models, EfficientNet-B0 showed the highest accuracy, improving upon the results obtained from the previous study using Inception-ResNet-v2 with wrist bones in the MURA set (Ac = 0.869, κ = 0.728). The superior accuracy of EfficientNet-B0 (Ac = 0.869, κ = 0.732) also suggests that the EfficientNet model is more efficient than the Inception-ResNet-v2 model, given that it has approximately 1/10 of the parameters. Further, by integrating the HyperColumn-CBAM structure into the EfficientNet-B0 and DenseNet169 models, enhancements were observed in both models. The HyperColumn-CBAM-DenseNet169 model showcased the most superior performance with an accuracy of 0.875 and a κ value of 0.746.

Utilizing the CBAM-DenseNet169 model, a profound improvement in prediction accuracy was observed upon integrating the GRAZPEDWRI-DX dataset. Characterized by its superior image quality and expansive volume, the inclusion of the GRAZPEDWRI-DX dataset led to a significant accuracy leap, reaching an impressive 0.968. While the MURA dataset included voice samples, the GRAZPEDWRI-DX dataset did not. Nevertheless, the enhanced image quality, superior resolution, and larger sample size of the GRAZPEDWRI-DX dataset played pivotal roles in the notable performance enhancement observed. Such attributes of the GRAZPEDWRI-DX dataset can offer a distinct advantage for CNN, potentially leading to further improvements in accuracy.

Table 4 presents a summary of various studies focusing on wrist fracture detection, detailing the models and their results. It is important to note that a direct comparison between some of the studies, such as the work by Kim et al. and Lindsey et al., is challenging due to the utilization of different datasets. For instance, while Ananda et al. conducted experiments using the MURA dataset and achieved an accuracy of 86.9%, our proposed method demonstrated a slight improvement, achieving 87.5%. In another instance, the method by Nagy et al., which used the GRAZPEDWRI-DX dataset, reported a precision of 0.917. In contrast, our proposed method, when tested on the same dataset, achieved a precision of 0.968.

Utilizing our Grad-CAM-based color-visualization technique, we identified misalignments in the heatmap produced by the DenseNet169 model. More precisely, the heatmap either mislocalized regions resembling fractures or was overly broad in its depiction. However, when we transitioned to the HyperColumn-CBAM-DenseNet169 model, there was a marked improvement in the Mean of Intersection over Union (MoI), elevating from a mere 0.083 to an impressive 0.213. This can be visualized in Figure 11.

This shift in IoU is crucial. An IoU of 0.083, in essence, means that there is an overwhelming 87% misalignment, making predictions practically non-actionable. An IoU of 0.213, while still hinting at a 40% misalignment, showcases a significant enhancement, edging closer to a more accurate level of localization. Such progress implies that although the information from the final layer might be too generalized for pinpoint location accuracy, the HyperColumn-CBAM model excels by leveraging spatial details from preceding layers. Consequently, the outcomes align more closely with the expert-identified fracture zones, underscoring its potential as an invaluable tool in fracture diagnosis.

As shown in Figure 10, the model with the coordinates added further aligned with the fracture as marked by the surgeon. Despite these advantages, the object-detection deep-learning architecture is known to require a considerable volume of labeled bounding box training data. In order to detect the presence or absence of fractures and abnormalities in the upper extremity, a vast amount of labeled bounding box data, including fractures, implants, and tissue abnormalities, is required. However, the time of an imaging specialist is valuable, and labeling these disease areas is a time-intensive task. Yet, our proposed model, even when trained only with fracture/non-fracture labeled data without bounding boxes, aligns closely with the fracture site marked by the expert. This makes it highly suitable for large-scale training, offering the advantage of improving training and diagnostic accuracy as acquiring a large volume of surrogate data is relatively easy.

A limitation in this study is acknowledged. While the model effectively detects the primary fracture site, it does not differentiate between multiple fractures within the same image. This limitation stems from the binary training strategy, which only discerns the presence or absence of fractures. Consequently, the model is predominantly oriented towards detecting the most prominent fracture site. To harness the full potential of the model and broaden its detection capabilities, subsequent research should emphasize the identification of multiple fractures and other irregularities.

## 7. Conclusions

In this study, ten distinct models were utilized to identify and classify fractures within wrist X-ray images. The introduction of the HyperColumn-CBAM structures into the EfficientNet-B0 and DenseNet169 models marked a significant leap in accurately predicting fracture sites. This allowed for a more concentrated heatmap representation, which impressively mirrored the region pinpointed by the medical expert.

A cornerstone of our methodology is the adoption of a model anchored solely on fracture/non-fracture labeled data, eliminating the conventional reliance on bounding box annotations. Such a pivot effectively circumvents the major roadblock associated with object detection in deep-learning architectures that historically hinge on extensive labeled bounding box datasets. The outcome is a model that aligns seamlessly with the regions identified by experts, streamlining the training process and establishing a foundation for improved diagnostic accuracy.

Our findings not only pave the way for future advancements in diagnostic tools but also hint at the prospect of harnessing larger and more varied datasets to refine predictions and outcomes further. This study highlights a promising direction that provides considerable time and financial efficiencies in radiological fracture identification. Such innovations hold immense promise, especially for professionals in high-pressure domains like emergency medicine, ultimately setting the stage for enhanced patient care and prognosis.

## Figures and Tables

**Figure 1 diagnostics-13-02927-f001:**
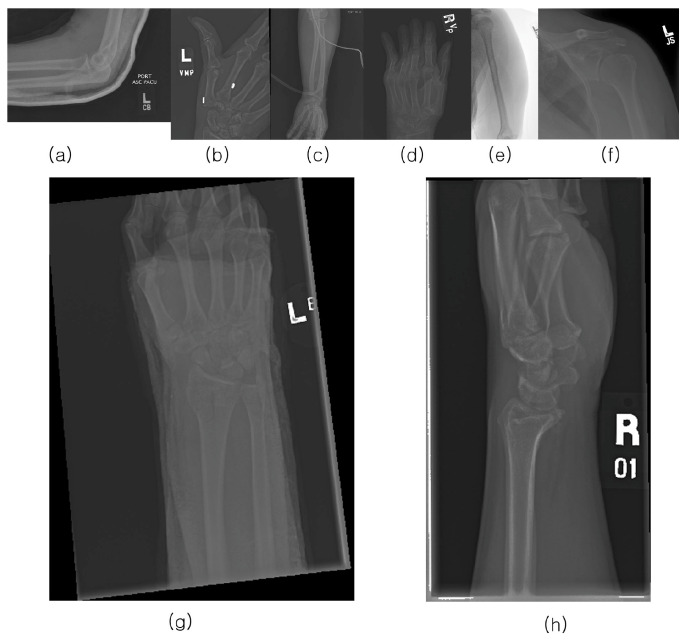
Positive MURA dataset: (**a**) elbow, (**b**) finger, (**c**) forearm, (**d**) hand, (**e**) humerus, (**f**) shoulder, (**g**) wrist, (**h**) wrist.

**Figure 2 diagnostics-13-02927-f002:**
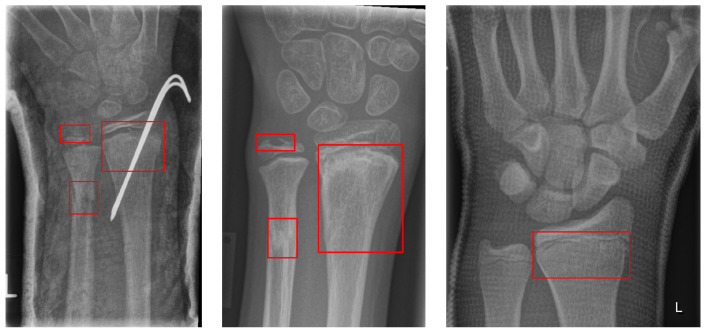
GRAZPEDWRI-DX dataset with the red box indicating the ground truth fracture region.

**Figure 3 diagnostics-13-02927-f003:**
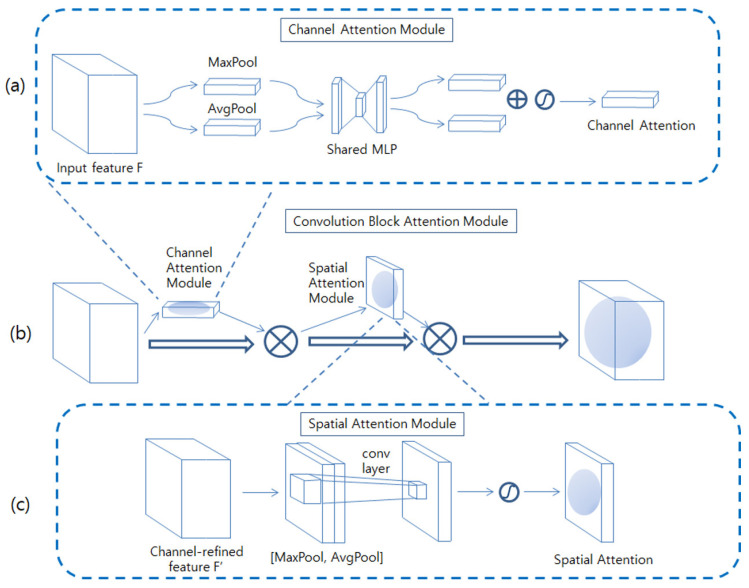
(**a**) Channel attention module, (**b**) convolutional block attention module, (**c**) spatial attention module.

**Figure 4 diagnostics-13-02927-f004:**
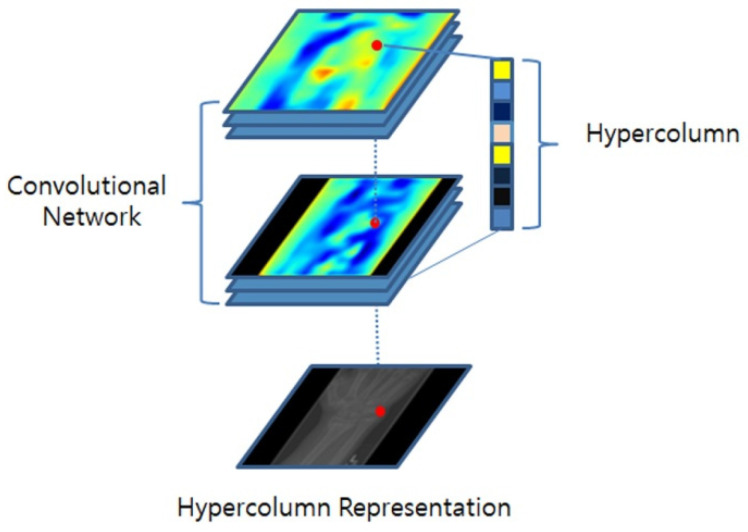
HyperColumn conceptual diagram.

**Figure 5 diagnostics-13-02927-f005:**
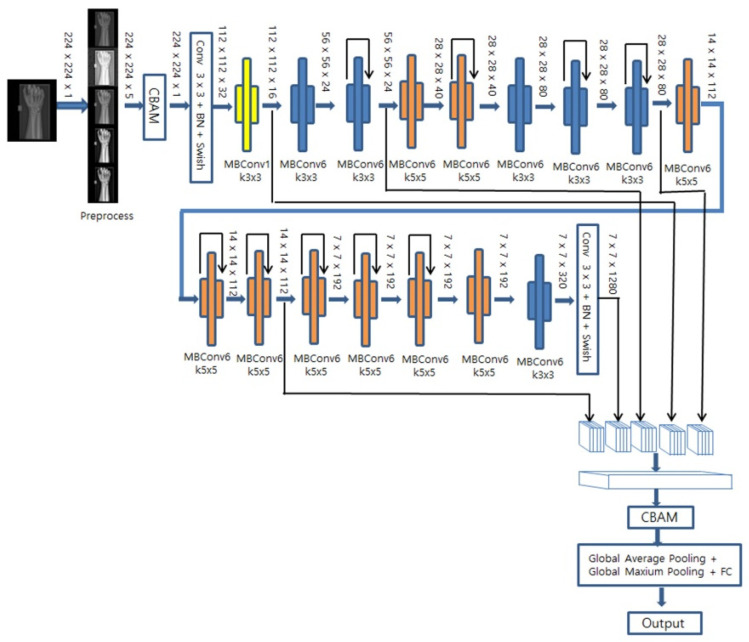
EfficientNet-B0 base HyperColumn-CBAM model.

**Figure 6 diagnostics-13-02927-f006:**
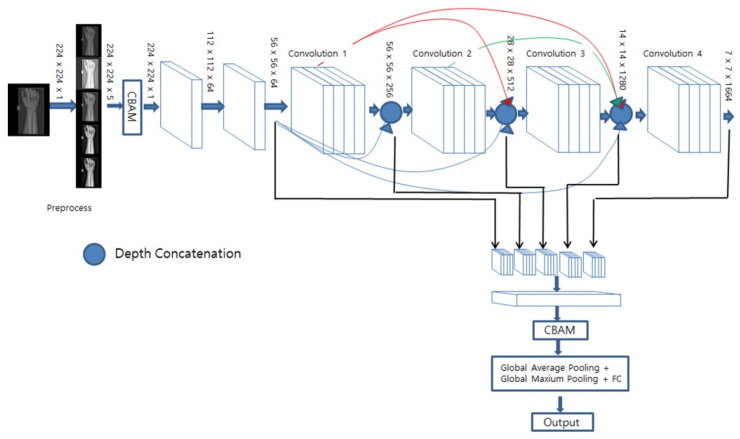
Densenet169 base HyperColumn-CBAM model.

**Figure 7 diagnostics-13-02927-f007:**
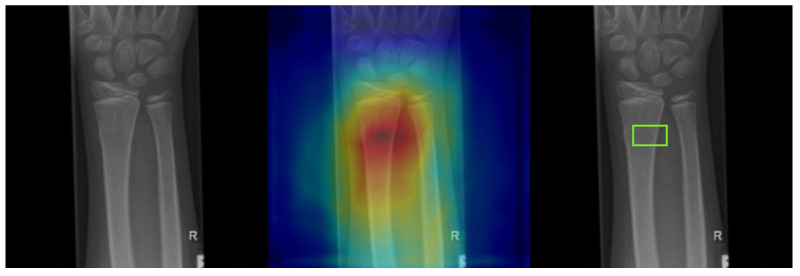
Heat-map extracted from the HyperColumn-CBAM-Densenet169 model, with the green box indicating the ground truth fracture region.

**Figure 8 diagnostics-13-02927-f008:**
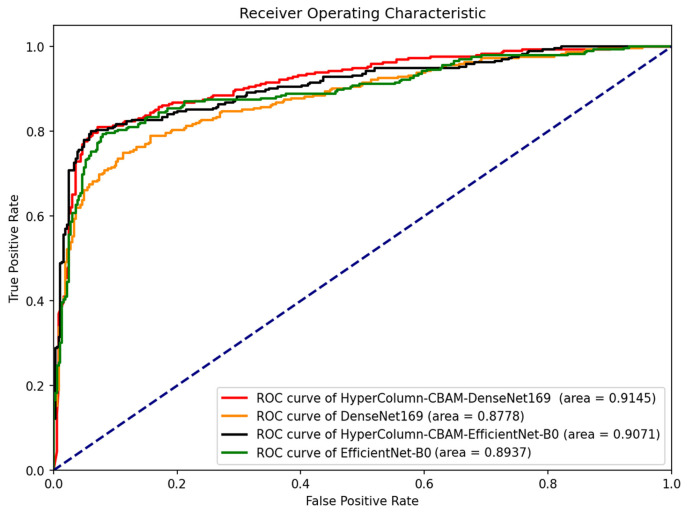
Classification results of the CNN models, with and without HyperColumn-CBAM enhancement.

**Figure 9 diagnostics-13-02927-f009:**
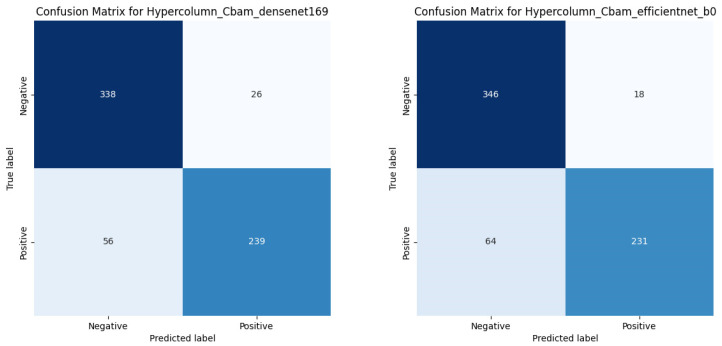
Confusion matrix of the HyperColumn-CBAM-Densenet169 and the HyperColumn-CBAM-EfficientNet-B0.

**Figure 10 diagnostics-13-02927-f010:**
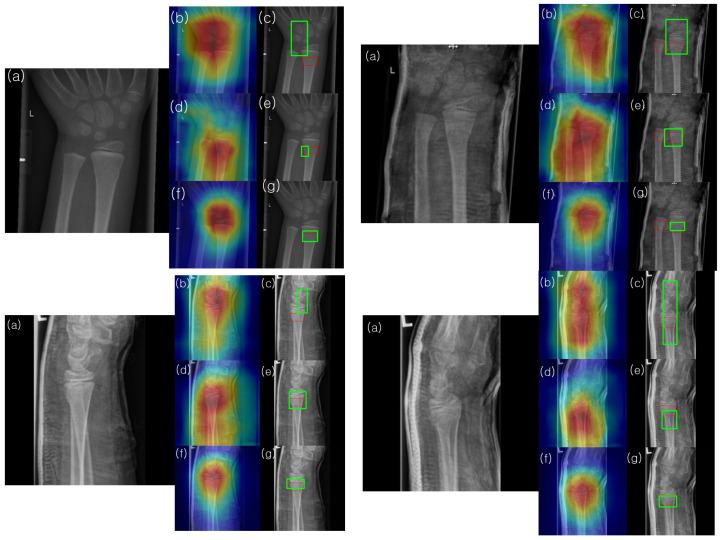
Illustration of the activated image regions for the models in Table 3: (**a**) input X-ray image, (**b**) overlay of the input image with the activated heatmap of the DenseNet169 model, (**c**) predicted square (green) and ground truth squares (red) obtained by thresholding the activated heatmap of the DenseNet169 model, (**d**) overlay of the input image with the activated heatmap of the HyperColumn-DenseNet169 model, (**e**) predicted square (green) and ground truth (red) obtained by thresholding the activated heatmap of the HyperColumn-DenseNet169 model, (**f**) overlay of the input image with the heatmap of the HyperColumn-DenseNet169 model trained by adding coordinate information, (**g**) predicted square (green) and ground truth (red) obtained by thresholding the activated heatmap of the HyperColumn-DenseNet169 model trained by adding coordinate information.

**Figure 11 diagnostics-13-02927-f011:**
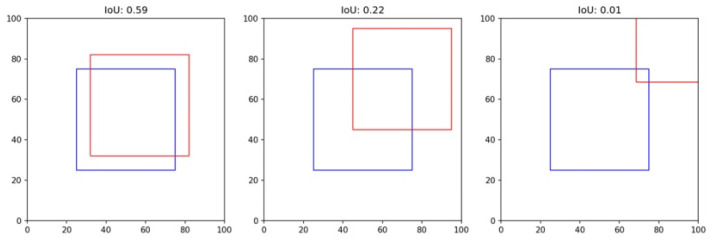
Example of misalignment and IoU of two boxes (blue is ground truth, red is predicted).

**Table 1 diagnostics-13-02927-t001:** Overview of the deep-learning network models used.

Model	Depth	Input Size
VGG19	16 convolutional layers + 3 fully connected layers	224 × 224
ResNet50	50 layers (49 convolutional + 1 fully connected layer) with 4 residual blocks	224 × 224
Inception V3	48 convolutional + 1 global average pooling layer using factorized convolutions	299 × 299
Xception	71 convolutional + 1 global average pooling layer with inception module using depth-wise separable convolutions	299 × 299
DenseNet169	169 convolutional + 1 global average pooling + 1 fully connected layer with densely connected block	224 × 224
EfficientNet-B0	18 convolutional + 1 global average pooling layer + fully connected layer with mobile inverted bottleneck convolution	224 × 224
RegNetY-064	64 stages (sequence of convolutional layers)	224 × 224
NASNet-A Mobile	Complex structure with around 1056 operations and 4 auxiliary heads	224 × 224
Vit_base_patch16_224	12 transformer layers	224 × 224
Swin Transformer Small	28 layers	224 × 224

**Table 2 diagnostics-13-02927-t002:** Classification results for the wrist in the MURA dataset. The best results are highlighted in bold.

Model	Accuracy	Precision	Recall	F1	k
VGG19	0.844	0.854	0.837	0.838	0.678
ResNet50	0.857	0.881	0.843	0.852	0.704
Inception v3	0.856	0.867	0.846	0.851	0.703
Xception	0.857	0.871	0.847	0.852	0.706
DenseNet169	0.859	0.867	0.850	0.854	0.710
EfficientNet-B0	0.869	0.876	0.861	0.866	0.732
RegNetY-064	0.853	0.872	0.840	0.850	0.695
NASNet-A Mobile	0.852	0.866	0.840	0.845	0.693
Vit_base_patch16_224	0.857	0.875	0.846	0.852	0.705
Swin Transformer Small	0.852	0.863	0.843	0.848	0.697
HyperColumn-CBAM Model-DenseNet169	**0.875**	0.880	**0.869**	**0.873**	**0.746**
HyperColumn-CBAM Model-EfficientNet-B0	0.875	**0.885**	0.867	0.872	0.744

**Table 3 diagnostics-13-02927-t003:** Results of classifying the wrist in the MURA dataset and GRAZPEDWRI-DX. The best results are highlighted in bold.

Model	Accuracy	Precision	Recall	F1	k	IoU
DenseNet169	0.946	0.921	0.937	0.929	0.857	0.083
HyperColumn-CBAM Model-DenseNet169	**0.968**	**0.955**	**0.958**	**0.957**	**0.913**	0.213
HyperColumn-CBAM Model-DenseNet169 with coordinate	0.947	0.923	0.936	0.929	0.859	**0.574**

**Table 4 diagnostics-13-02927-t004:** Comparative studies on wrist-fracture-detection methods.

Studies	Methods Used	Dataset	Result
Kim et al. [21]	Inception V3	Royal Devon and Exeter Hospital	0.954(AUROC)
Lindsey et al. [22]	U-Net	Hospital for Special Surgery	0.967(AUC)
Ananda et al. [23]	Inception-ResNet-V2	MURA	0.869(Ac)
Nagy et al. [24]	Yolo5	GRAZPEDWRI-DX	0.917(Precision)
Ju et al. [25]	YOLOv8	GRAZPEDWRI-DX	0.629(mAP 50)
Developed architecture	HyperColumn-CBAM Model-DenseNet169	MURA	0.875(Ac)
Developed architecture	HyperColumn-CBAM Model-DenseNet169	MURA and GRAZPEDWRI-DX	0.968(Ac) 0.955(Precision)

## Data Availability

Data used in this study are available at https://stanfordmlgroup.github.io/competitions/mura/ (accessed on 7 March 2023) and https://www.nature.com/articles/s41597-022-01328-z (accessed on 31 May 2023).

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
