# Peer review of "Enhancing X-ray-Based Wrist Fracture Diagnosis Using HyperColumn-Convolutional Block Attention Module"

_diagnostics, 2023, doi:10.3390/diagnostics13182927_

Round 1
Reviewer 1 Report
Some comments given to authors below:
1. Line 27, What is the current article novel? It has been extensively discussed in the past. Nothing truly novel in its current state. The absence of anything original makes the current study seem like a replication or a modified study. The introduction section should contain specifics about the writers' uniqueness. It is a significant reason to reject this study.
2. Line 28, suggested to using more updated data from 2023 for patient fracture.
3. Line 33, the proven and rationalisation that x-ray imaging has been being used in old century should be given.
4. Line 125, what is the pocus in point of view from illustrated fracture? Figure 2 not interactive and suggested to improve the presentation.
5. Line 150, giving comprehensive discussion regarding Overview of the deep learning network models used, not just mentioned it.
6. Figure 3-5 is similar, and suggested to combine it with splitting in part a, b, and c frather than different figure.
7. Please explain potential further study performing computational simulation/in silico. It bring several advantages compared to in vivo/clinical and in vitro/laboratory study such as lower cost and faster results. For this purpose, please provide the explanation along with relevant reference as follows: https://doi.org/10.3390/biomedicines11030951
-
Author Response
Dear Editor and Reviewers,
We express our sincere gratitude to both the editorial team and the reviewers for dedicating time to meticulously review our manuscript and offering invaluable feedback. We have taken all comments into consideration and endeavored to address each one comprehensively. To ensure the clarity and fluency of our manuscript, we also sought the English correction service provided by MDPI. We hope the manuscript has been improved accordingly.
All modifications in the manuscript have been marked up using the “Track Changes” function. Below we provide the point by point responses.
We deeply value your feedback and are hopeful that our revisions meet your expectations. We eagerly await your further thoughts on our revised manuscript and remain open to any additional feedback to enhance its quality.
Sincerely,
Joong Lee
Response to Reviewer 1
[Comment 1] Line 27, What is the current article novel? It has been extensively discussed in the past. Nothing truly novel in its current state. The absence of anything original makes the current study seem like a replication or a modified study. The introduction section should contain specifics about the writers' uniqueness. It is a significant reason to reject this study.
Response: We appreciate your feedback. The introduction has been revised and enhanced to better emphasize the unique contributions and novel aspects of this research.
[Comment 2] Line 28, suggested to using more updated data from 2023 for patient fracture.
Response: Revised accordingly.
[Comment 3] Line 33, the proven and rationalisation that x-ray imaging has been being used in old century should be given.
Response: We appreciate your feedback. We have now provided the specific year and the name of the hospital where X-ray imaging was first introduced to substantiate our claim.
[Comment 4] Line 125, what is the pocus in point of view from illustrated fracture? Figure 2 not interactive and suggested to improve the presentation.
Response: We have revisited Figure 2 and made necessary adjustments to better emphasize the focus and improve its clarity
[Comment 5] Line 150, giving comprehensive discussion regarding Overview of the deep learning network models used, not just mentioned it.
Response: We appreciate your insightful suggestion. We have expanded our discussion on the deep learning network models used to provide a more comprehensive understanding.
[Comment 6] Figure 3-5 is similar, and suggested to combine it with splitting in part a, b, and c frather than different figure.
Response: Thank you for the suggestion. We have combined Figures 3-5 into a single figure with parts a, b, and c for clarity and conciseness.
[Comment 7] Please explain potential further study performing computational simulation/in silico. It bring several advantages compared to in vivo/clinical and in vitro/laboratory study such as lower cost and faster results. For this purpose, please provide the explanation along with relevant reference as follows: https://doi.org/10.3390/biomedicines11030951.
Response: Revised accordingly
Reviewer 2 Report
I have examined your work titled"Developing X-ray Based Wrist Fracture Diagnosis Using Hyper-2 Column CBAM" in detail. I have listed the missing points in the article in detail. In the abstract, the purpose of the study and the innovations of the study should be highlighted. The Releated work section should be expanded with similar studies. A paragraph about the organization of the article should be added at the end of the Introduction section. Innovations and the importance of the model should be emphasized. Although many models were used in the study, the Result section was insufficient. The relevant section should be supported, such as confusion matrix, etc. Limitations of the study should be included and related studies should be presented in a table so that it would facilitate interpretation. Adressing a shorcomings I mentioned in thr discussion section will carry the study on the next segment.
.
Author Response
Dear Editor and Reviewers,
We express our sincere gratitude to both the editorial team and the reviewers for dedicating time to meticulously review our manuscript and offering invaluable feedback. We have taken all comments into consideration and endeavored to address each one comprehensively. To ensure the clarity and fluency of our manuscript, we also sought the English correction service provided by MDPI. We hope the manuscript has been improved accordingly.
All modifications in the manuscript have been marked up using the “Track Changes” function. Below we provide the point by point responses.
We deeply value your feedback and are hopeful that our revisions meet your expectations. We eagerly await your further thoughts on our revised manuscript and remain open to any additional feedback to enhance its quality.
Sincerely,
Joong Lee
Response to Reviewer 2
[Comment 1] In the abstract, the purpose of the study and the innovations of the study should be highlighted.
Response: We appreciate your feedback. We have updated the abstract to clearly state the study's purpose and innovations introduced in this research..
[Comment 2] The Releated work section should be expanded with similar studies.
Response: We have expanded the "Related Work" section to incorporate more relevant studies.
[Comment 3] A paragraph about the organization of the article should be added at the end of the Introduction section.
Response: We've made the change accordingly.
[Comment 4] Innovations and the importance of the model should be emphasized. Although many models were used in the study, the Result section was insufficient. The relevant section should be supported, such as confusion matrix, etc.
Response: Thank you for highlighting the importance of emphasizing our model's innovation and significance. We revisited the Results section and augmented it with a detailed explanation, including the addition of a confusion matrix, to provide a clearer understanding of the model's performance. Furthermore, in the Discussion section, we accentuated the distinctiveness of our methodology and its implications for the field.
[Comment 5] Limitations of the study should be included and related studies should be presented in a table so that it would facilitate interpretation. Adressing a shorcomings.
Response: Thank you for your suggestion. We have incorporated a section detailing the limitations of our study in the Discussion section.
Reviewer 3 Report
-The paper proposes a ML-based algorithms for wrist fracture diagnosis aid.
-The approach appears valid, but the methodology and especially the manuscript, should be improved in particular:
- authors should favour the use of the 3-rd person rather than "we";
- when using references, there should be a space between text and "[" ;
- when quoting other works (eg: in line 47, Andre Esteva et al) a reference number should be provided;
- a reference to (possible) past works on HyperColumn (line 55) should be considered;
- In line 106, it should be clarified if the number of images of the validation dataset is only for wrist images or in total;
- In figure 1, in the present form it is hard to identify the lesion region;
- It is quoted that only the MURA datse is used (lins 127-128), but results for GRAZPEDWRI-DX are alos presented (eg: Table 3)... please clarify;
- In Eq. 3, the numerator should be (TP+TN) (parenthesis missing);
- A reference to Cohen's kappa (eq. 7), as done with other metrics, would help on readability;
- In table 2, Cohen's kappa can be simply K, as previously introduced;
- The performance in terms of IoU (table 3) are really far too modest, urging for a deeper analysis.
(comments and suggestions above)
Author Response
Dear Editor and Reviewers,
We express our sincere gratitude to both the editorial team and the reviewers for dedicating time to meticulously review our manuscript and offering invaluable feedback. We have taken all comments into consideration and endeavored to address each one comprehensively. To ensure the clarity and fluency of our manuscript, we also sought the English correction service provided by MDPI. We hope the manuscript has been improved accordingly.
All modifications in the manuscript have been marked up using the “Track Changes” function. Below we provide the point by point responses.
We deeply value your feedback and are hopeful that our revisions meet your expectations. We eagerly await your further thoughts on our revised manuscript and remain open to any additional feedback to enhance its quality.
Sincerely,
Joong Lee
Response to Reviewer 3
[Comment 1] authors should favour the use of the 3-rd person rather than "we"
Response: Changes have been made throughout the manuscript to adhere to this recommendation.
[Comment 2] when using references, there should be a space between text and "[".
Response: Adjustments have been made throughout the paper to ensure proper spacing.
[Comment 3] when quoting other works (eg: in line 47, Andre Esteva et al) a reference number should be provided.
Response: We have revisited the entire document and ensured that all references are properly cited, especially at line 47 where Andre Esteva et al. are mentioned.
[Comment 4] a reference to (possible) past works on HyperColumn (line 55) should be considered.
Response: References related to HyperColumn have been added as your suggestion.
[Comment 5] In line 106, it should be clarified if the number of images of the validation dataset is only for wrist images or in total.
Response: Clarifications have been made in line 106 to specify the nature of the images in the validation dataset.
[Comment 6] In figure 1, in the present form it is hard to identify the lesion region.
Response: We've enhanced Figure 1 to make the lesion region more distinguishable..
[Comment 7] It is quoted that only the MURA datse is used (lins 127-128), but results for GRAZPEDWRI-DX are alos presented (eg: Table 3)... please clarify.
Response: We've made the change accordingly.
[Comment 8] In Eq. 3, the numerator should be (TP+TN) (parenthesis missing)
Response: The error in Eq. 3 has been rectified.
[Comment 9] In table 2, Cohen's kappa can be simply K, as previously introduced.
Response: We've made the change accordingly.
[Comment 10] The performance in terms of IoU (table 3) are really far too modest, urging for a deeper analysis.
Response: Thank you for your feedback on the IoU performance as presented in Table 3. We conducted a more in-depth analysis and emphasized the importance of IoU as a metric for our study. While the absolute values might seem modest, it's essential to highlight the relative improvement and its implications. An increase in IoU, even if incremental, represents a significant enhancement in localization accuracy, which is pivotal for medical imaging diagnostics.
Reviewer 4 Report
This paper is an empirical study report on automated X-ray image classification for wrist fracture detection. A series of convolutional neural networks are considered for model building. Hybrid attention built with Spatial and Channel also was incorporated for performance boosting. The performances are compared with several statistics. GradCAM-based explainability is applied finally. The effort available here makes me inclined to recommend this paper for acceptance. However, the development or advancement of the chosen algorithm to fill the performance gap was good. Additionally, the author can try to put the key findings related to this application.
Fine. Some minor mistakes are there.
Author Response
Dear Editor and Reviewers,
We express our sincere gratitude to both the editorial team and the reviewers for dedicating time to meticulously review our manuscript and offering invaluable feedback. We have taken all comments into consideration and endeavored to address each one comprehensively. To ensure the clarity and fluency of our manuscript, we also sought the English correction service provided by MDPI. We hope the manuscript has been improved accordingly.
All modifications in the manuscript have been marked up using the “Track Changes” function. Below we provide the point by point responses.
We deeply value your feedback and are hopeful that our revisions meet your expectations. We eagerly await your further thoughts on our revised manuscript and remain open to any additional feedback to enhance its quality.
Sincerely,
Joong Lee
Response to Reviewer 4
[General Comment] This paper is an empirical study report on automated X-ray image classification for wrist fracture detection. A series of convolutional neural networks are considered for model building. Hybrid attention built with Spatial and Channel also was incorporated for performance boosting. The performances are compared with several statistics. GradCAM-based explainability is applied finally. The effort available here makes me inclined to recommend this paper for acceptance. However, the development or advancement of the chosen algorithm to fill the performance gap was good. Additionally, the author can try to put the key findings related to this application.
Response: Thank you for your positive feedback and your encouraging comments. We have worked on elaborating the advancements of our chosen algorithm to clearly illustrate its impact on bridging the performance gap. Furthermore, we've incorporated a confusion matrix to provide clearer insights into the model's efficacy. Your insights have been invaluable in refining our manuscript.
Round 2
Reviewer 2 Report
Thank you for correcting the shortcomings I have mentioned in the revision. In the Abstract section, please add information about the results of the study. Our, we etc. Avoid using words. Add a table with the related work in the Discussion section. In this table, the studies done, the methods used, the results obtained, etc. include the results. You can review the related study "Automatic classification and diagnosis of heart valve diseases using heart sounds with MFCC and proposed deep model". Reference should be added to the Evaluation Metrics section. The Discussion section looks more like the Introdution section. It is possible to increase the number of studies examined in the Related work section. In conclusion, thank you again for addressing many shortcomings in the previous revision. Eliminating the deficiencies that are overlooked in the studies will increase the quality of the studies.
It is important to eliminate spelling and grammatical errors and to avoid words like we, our whenever possible.
Author Response
Dear Reviewer,
We express our sincere gratitude to reviewer for dedicating time to meticulously review our manuscript and offering invaluable feedback. We have taken all comments into consideration and endeavored to address each one comprehensively. We hope the manuscript has been improved accordingly.
All modifications in the manuscript have been marked up using red color. Below we provide the point by point responses.
We deeply value your feedback and are hopeful that our revisions meet your expectations. We eagerly await your further thoughts on our revised manuscript and remain open to any additional feedback to enhance its quality.
Sincerely,
Joong Lee
Response to Reviewer
[Comment 1] In the Abstract section, please add information about the results of the study. Our, we etc. Avoid using words.
Response: Thank you for pointing this out. I have revised the Abstract section to include the results of the study and ensured that we refrained from using first-person pronouns..
[Comment 2] Add a table with the related work in the Discussion section. In this table, the studies done, the methods used, the results obtained, etc. include the results. The Discussion section looks more like the Introdution section.
Response: I appreciate the feedback. We have now included a table in the Discussion section that provides a comprehensive summary of related works, detailing the methodologies and results from each study. We have also revised the content of the Discussion section to differentiate it from the Introduction.
[Comment 3] You can review the related study "Automatic classification and diagnosis of heart valve diseases using heart sounds with MFCC and proposed deep model". It is possible to increase the number of studies examined in the Related work section.
Response: Thank you for the suggestion. We have reviewed and incorporated insights from the mentioned study into the Related Work section, providing a broader context and understanding of the field.
[Comment 4] Reference should be added to the Evaluation Metrics section.
Response: We have now added the necessary references to the Evaluation Metrics section to provide a clear foundation for our evaluation methods.
Reviewer 3 Report
The new version of the manuscript accommodates my previous comments
Author Response
Dear Reviewer,
Thank you for your time and effort in reviewing our manuscript and for your constructive feedback. We are gratified to learn that the revised version of our manuscript meets your expectations and addresses the concerns and recommendations you provided in the prior review.
We value your input in this process and appreciate your continued consideration.
Sincerely,
Joong Lee
Response to Reviewer
[Comment] The new version of the manuscript accommodates my previous comments.
Response: We are grateful for your positive feedback and are pleased that our revisions have addressed your previous comments satisfactorily.
Round 3
Reviewer 2 Report
Thank you for correcting the deficiencies in the study in the revision. It is important that you review spelling and grammatical errors. For example, "EfficientNet", densenet169 is spelled differently in different places. Attention should be paid to uppercase and lowercase letters.
It is important that you review spelling and grammatical errors
Author Response
Dear Reviewer,
Thank you for your careful consideration and feedback on our manuscript. We greatly appreciate the time and effort you've dedicated to enhancing the quality of our work.
Below, we address the specific comments you provided.
[Comment] Thank you for correcting the deficiencies in the study in the revision. It is important that you review spelling and grammatical errors. For example, "EfficientNet", densenet169 is spelled differently in different places. Attention should be paid to uppercase and lowercase letters.
Response: Apologies for the oversight. The inconsistencies in the naming of "EfficientNet" and "DenseNet169" have been corrected throughout the manuscript. Additionally, to ensure the quality and clarity of the manuscript, it has undergone a professional English language editing service provided by a MDPI English service. We trust this will address any lingering concerns regarding the linguistic quality of the text.
We are hopeful that these corrections meet your expectations. Once again, we value your feedback and will strive to maintain the quality and accuracy of our work.
Sincerely,
Joong Lee